# Therapeutic itineraries of snakebite victims and antivenom access in southern Mexico

**Chloe Vasquez[1], Edgar Neri Castro [2], Eric D. Carter [3] ***

**1** Macalester College, Saint Paul, Minnesota United States of America, **2** Facultad de Ciencias Biológicas, Universidad Juárez del Estado de Durango, Durango, Mexico, **3** Macalester College, Saint Paul, Minnesota United States of America

☙ These authors contributed equally to this work.

* ecarter@macalester.edu

## Abstract

Access to antivenoms in cases of snakebite continues to be an important public health issue around the world, especially in rural areas with poorly developed health care systems. This study aims to evaluate therapeutic itineraries and antivenom accessibility following snakebites in the states of Oaxaca and Chiapas in southern Mexico. Employing an intercultural health approach that seeks to understand and bridge allopathic and traditional medical perceptions and practices, we conducted field interviews with 47 snakebite victims, documenting the therapeutic itineraries of 54 separate snakebite incidents that occurred between 1977 and 2023. Most victims used traditional remedies as a first line of treatment, often to withstand the rigors of a long journey to find antivenoms. The main obstacles to antivenom access were distance, poor antivenom availability, and cost. Standard antivenom treatment is highly valued and sought after, even as traditional beliefs and practices persist within a cultural framework known as the "hot-cold" system. The findings are crucial for informing future enhancements to antivenom distribution systems, health education initiatives, and other interventions aimed at mitigating the impact of snakebites in the region.

## Author summary

Every year, hundreds of thousands of people worldwide are killed, disabled, or disfigured by bites from venomous snakes. Snakebite is especially dangerous in rural areas of the global south, where victims often must travel long distances to clinics or hospitals to receive antivenom treatment. What sorts of obstacles do snakebite victims face in their search for antivenom? And why do people choose traditional remedies, such as homeopathic treatments, even though these have limited evidence of effectiveness against snakebite envenoming? To answer these questions, we conducted field research in rural areas of Oaxaca and Chiapas states in southern Mexico in 2023, including interviews with 47 snakebite victims. In line with other studies, we found that the main obstacles to antivenom access are distance, cost, and scarcity of antivenom supply. And by listening more closely to victims' accounts of their therapeutic itineraries, we have a better understanding of why people might choose traditional remedies for snakebite. These reasons include

**Data Availability Statement:** All data are in the manuscript and/or supporting information files.

**Funding:** CV received funding (as a US$6000 stipend) for this project from Macalester College through a Mann-Hill Summer Research Fellowship

(no grant number available). The funders had no role in study design, data collection and analysis, decision to publish, or preparation of the manuscript. (https://www.macalester.edu/olin-rice-hub/research-opportunities/wintermannhill/).

**Competing interests:** The authors have declared that no competing interests exist.

long-established cultural frameworks about the causes of illness and the need to alleviate pain to withstand the long journey to conventional treatment. By taking an intercultural approach, which emphasizes mutual respect and understanding between Western and indigenous health belief systems, we can improve access to antivenom, health education initiatives, and other interventions aimed at mitigating the impact of snakebites in Mexico and beyond.

## Introduction

Former United Nations secretary general Kofi Annan described snakebite as "the biggest public health crisis you've never heard of" [1]. Despite the dearth of attention and response from global health actors, snakebite envenoming (SBE) poses a major threat to safety and wellbeing in rural populations across the globe. Snakebites can cause a variety of symptoms, ranging from pain and swelling to hemorrhage, necrosis, paralysis, and death [2]. An estimated 5.4 million people are bitten annually, with up to 2.7 million people experiencing envenoming; 81,000–138,000 people die from SBE each year and another 400,000 experience permanent disability [3]. SBE is a disease of the poor, most prevalent among agricultural workers, fishers, hunters, indigenous people, and rural dwellers with little access to health or educational resources [4]. An estimated 95% of SBE cases occur in tropical and/or developing countries [5], where healthcare systems are often underfunded and overwhelmed, pharmaceutical regulation is weak, and antivenom distribution is fragmented and inconsistent. Timely antivenom access is associated with lower rates of mortality, tissue and limb loss, and liver damage [6]; however, providing antivenom to snakebite victims, especially in rural areas of low-to-middle income countries, can be a challenge.

There are three major barriers to antivenom access. First, geography: victims must often travel long distances to access even primary care facilities. Many have limited access to transportation or must use multiple methods of travel over difficult terrain to reach the clinic or hospital [7]. Second, antivenoms and the qualified staff who could administer them are unavailable in most rural clinics [8]. Third, costs: victims often face financial barriers to antivenom access, especially when public clinics and hospitals do not have it in stock. High commercial prices can contribute to under-dosing or use of traditional treatments as a substitute [8,9]. In addition to the factors above, other barriers to antivenom access include cultural norms, cold-chain limitations, and usage restrictions [3].

A growing literature analyzes the health-seeking behaviors of snakebite victims. Given the obstacles to accessing antivenom, there is often a gap between official recommendations of best therapeutic practices and how snakebite victims actually seek treatment. In practice, when individuals experience a venomous snakebite, they have a variety of treatment options. They could go to a clinic or a hospital, consult a traditional healer, use homeopathic remedies, or some combination of all of these. Often patients only choose to seek healthcare following complications or the onset of severe symptoms following a snakebite [7]. Some laboratory studies have shown that medicinal plants may act as enzyme inhibitors, but field studies are limited and effective dosages have not been determined [2,10,11]. While scientific research on the efficacy of medicinal plants for treating snakebite is ambiguous [12,13], homeopathic and herbal remedies are commonly used to treat snakebite around the world [14–16].

Anthropological research sheds light on such cultural practices and the principles behind them. In Mexico, along with other parts of the Meso-American region, rural beliefs and practices around snakebite are often framed within the "hot-cold" system [17–20]. In this

framework, illness results from a disruption of equilibrium between hot and cold states in the body. Treatments aim to restore balance, such that cold ailments are treated with "hot" remedies, and vice versa [20]; plants, foods, medicines, and illnesses have hot (*caliente*) and cold (*frío*) or cool (*fresco*) properties; snakebite is viewed as "cold" and thus "hot" remedies are required [21]. Previous research in southern Mexico, around our study area, have documented snakebite healing practices that follow the logic of the "hot-cold" concept, including the use of steam baths and fire cupping (*ventosas*); application of "hot" cane alcohol, boiling water, or chilies to snakebite wounds; avoiding "cold" water and "hot" foods (eggs, pork, meat, fat or oil), depending somewhat on the time after the bite; and avoiding pregnant women, a folk belief about snakebite that has been documented in many parts of Latin America [18,19,21].

In our study, we track the therapeutic itineraries of snakebite victims; this concept refers not only to the physical movement across time and space of those seeking treatment [7], but also an analysis of the personal, religious, and cultural rationales behind health-related decisions [22–23]. To aid in the analysis of the rationales behind treatment-seeking behaviors, we draw on the intercultural health model, which respects parallel and complementary uses of indigenous and biomedical therapies [9,23]. A study in coastal Ecuador found that indigenous Tsáchila individuals decided between allopathic and indigenous/folk medicine depending on the nature and gravity of the health problem, the accessibility of each treatment option, confidence in the effectiveness of treatment, and degree of trust in practitioners [23]. In Pichátaro, Michoacán, anthropological studies of treatment choice identified mixed folk-biomedical models shaped by indigenous and Western influences [24–25]. Indigenous knowledge systems have been passed down through generations, and are varied and dynamic. *Curandero* means healer, and can refer to traditional medical practitioners that use herbs, massages, or spiritual methods to cure. Curanderos who use spiritual healing methods can be called *chamanes* (shamans), while healers that prescribe plant teas, tinctures or poultices can be referred to as *yerberos* [23,26]. Some healers have implemented biomedical drugs, technologies and understandings of health problems alongside traditional knowledge [9]. Outside of traditional health practitioners, individuals and family groups often engage in self-care processes which integrate traditional and biomedical knowledge [9].

This study aims to evaluate therapeutic itineraries and antivenom accessibility after snakebites in southern Mexico. Employing an intercultural health approach, we conducted field interviews with snakebite victims in rural areas of Oaxaca and Chiapas states, including individuals who sought medical attention at clinics, those who did not, and those who received antivenom outside of conventional biomedical health facilities. Additionally, we explore perceptions of both biomedical and traditional snakebite treatments, to better understand rationales for treatment-seeking and structural barriers to treatment for SBE. The findings are crucial for informing future enhancements to antivenom distribution systems, health education initiatives, and other interventions aimed at mitigating the impact of snakebites in the region.

## Methods

### Ethics statement

Ethical consent was obtained from the Institutional Review Board of Macalester College. The authors obtained oral informed consent from all participants after explaining the goals and procedures of the interviews, and the voluntary nature of participation. This study uses pseudonyms for individuals and communities in order to protect the identity of study participants.

## Study area

Our study took place in two culturally and geographically distinctive areas of Southern Mexico, the Sierra Madre region (Chiapas state) and the Chinantla Baja region (Oaxaca state) (Fig 1). The Sierra Madre region in Chiapas consists of a volcanic mountain range parallel to the Southern coast. At lower elevations, land is predominantly covered by low deciduous forests, followed by pine-oak forests and mountain rainforests at middle elevations and evergreen cloud forests at higher elevations. Much of the forest has been replaced by coffee plantations. Chiapas' biodiversity makes it an important place for snakebite envenoming [27]. In Chiapas, there are 113 registered species, including 14 vipers and 7 species from the Elapidae family. 71 snake species are described for the Sierra Madre del Sur, including 6 species from the Viperidae family: *Metlapilcoatlus occiduus*, *Bothriechis bicolor*, *Bothrops asper*, *Cerrophidion godmani*, *Crotalus simus*, and *Porthidium dunni*, and 3 from the Elapidae family: *Micrurus browni*, *M. latifasciatus*, and *M. nigrocintus* are present [28]. Around 30% of the population of Chiapas state speaks an indigenous language, mainly Tseltal or Tzotzil.

The Chinantla Baja region of Oaxaca is part of the northeastern Papaloapan region at the foothills of the Sierra Madre mountain range. As in Chiapas, Oaxaca's topography and climate have fostered rich biodiversity, including the highest density of venomous snake species in Mexico [29]. There are 166 registered species in Oaxaca, of which, for its Sierra Madre region,

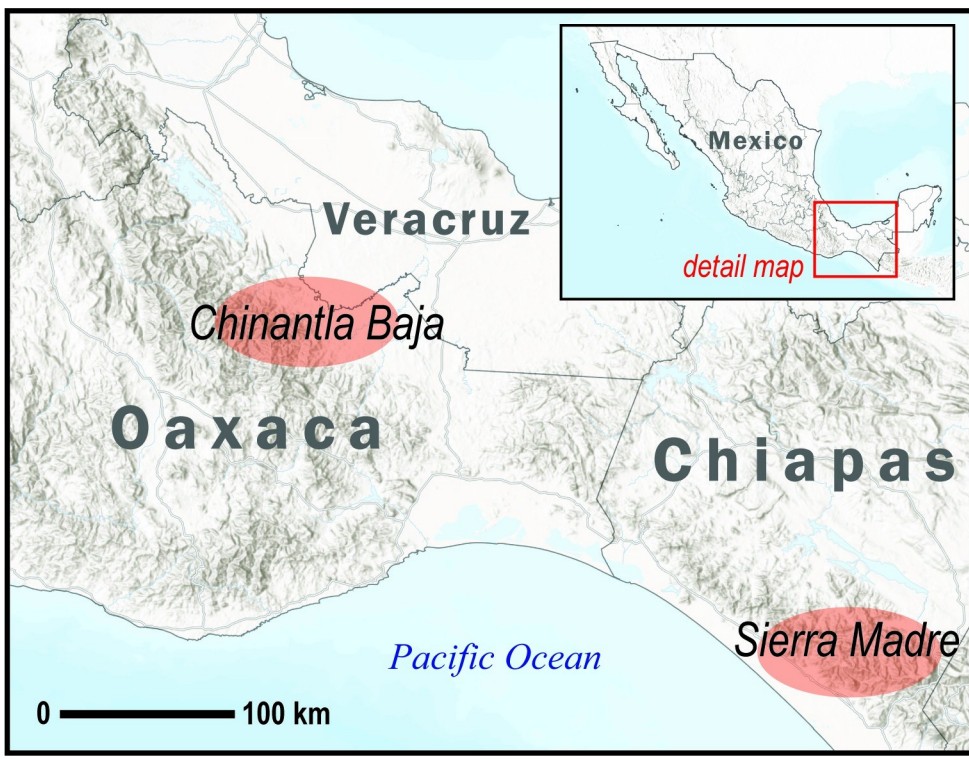

**Fig 1. Map of study area, by authors.** Created in ArcGIS and Adobe Illustrator software, using the following base layers: Mexico Country Boundary (https://services.arcgis.com/P3ePLMYs2RVChkJx/arcgis/rest/services/MEX_Boundaries_2022/FeatureServer), Mexico Estado Boundaries (https://services.arcgis.com/P3ePLMYs2RVChkJx/arcgis/rest/services/MEX_Boundaries_2022/FeatureServer), World Hillshade (https://services.arcgisonline.com/arcgis/rest/services/Elevation/World_Hillshade/MapServer), and World Terrain Base (https://cdn.arcgis.com/sharing/rest/content/items/33064a20de0c48d2bb61efa8faca93a8/resources/styles/root.json). There are no access and use limitations for these layers.

18 viper species have been recorded, including: *B. asper, C. petlalcalensis, Crotalus intermedius, C. molossus, Metlapilcoatlus nummifer, Ophryacus smaragdinus,* and *O. undulatus.* There are 7 registered species of elapids in Oaxaca, three of which can be found in the Sierra Madre: *M. diastema, M. elegans,* and *M. ephippifer* [30]. The most important species of venomous snake in the Chinantla Baja region are *Bothrops asper* (Sorda), *Metlapilcoatlus nummifer* (Mano de Metate), and *Micrurus sp.*, known as coral snakes. Oaxaca reports around 400 cases of snakebite envenoming annually [31], and most victims receive delayed care due to gaps in healthcare and communication. The population in the Chinantla Baja region is historically Chinantec, and about 14.5% of the population speaks an indigenous language [32].

## Study design

This study employed semi-structured interviews with open-ended questions to learn about snakebite victims' experiences and actions following their accident. Based on advice from local experts, the Sierra Madre (Chiapas) and Chinantla Baja (Oaxaca) were selected as study locations, due to their high snakebite incidence. With assistance from personnel of the National Commission of Protected Natural Areas (Comisión Nacional de Áreas Naturales Protegidas, or CONANP), we used snowball sampling in communities around El Triunfo Biological Reserve (Chiapas) to find 40 snakebite victims around the region. In Oaxaca, the 7 interviewees were found through conversations with fruit vendors and a local nurse, and through snowball sampling based on these contacts. Of the interviewed victims, some experienced multiple ophidic accidents. In total, the authors documented 54 snakebite incidents: 45 in the Sierra Madre region of Chiapas, and 9 in the Chinantla Baja region of Oaxaca. Statistics are calculated based on each case of snakebite, not by individual. Almost all the interviews were conducted with survivors of a snakebite incident, except for 2 interviews with surviving family members of deceased victims.

Interviews were conducted in Spanish, and lasted from 5 to 30 minutes. Notes were taken digitally. Field notes and interview transcripts were analyzed using Atlas.ti, a qualitative research software that allows for coding, thematic analysis, and data visualization. From this analysis, three themes emerged relating to barriers to antivenom: distance from medical facilities, unreliable antivenom distribution, and costs.

## Results

### Characteristics of snakebite incidents

Most interviewees were working-age males who had been bitten on farms, as well as in and around the house. All respondents described themselves as either mestizo, Tseltal, or Tzotzil (in Chiapas) or Chinantec (in Oaxaca). We documented 54 snakebite incidents, which took place between 1977 and 2023, among the 47 interviewees (Table 1). The most common anatomical site of the bite was the hand, followed by the leg (Fig 2). In 61% (n = 33) of incidents, traditional medicine was used, including use of herbal remedies (54%, n = 29), consulting with curanderos (28%, n = 15), consuming alcohol (37%, n = 20), and cutting and/or sucking the wound (24%, n = 13) (Fig 3). We did not consider tourniquet use as a traditional medicine method, as this used to be widely recommended by medical professionals until recently; tourniquets were used in 30% (n = 16) of incidents. In 21% (n = 11) of incidents, respondents reported following a special diet after the snakebite, composed mostly of broth, tostadas and black coffee, while excluding meat, fat, oil and other "hot" foods.

**Table 1. Characteristics of snakebite incidents.**

| Variables | Total (n = 54) | |
|---|---|---|
| | N | % |
| **Sex** | | |
| Male | 47 | 87% |
| **Age Group (years)** | | |
| 0–20 | 15 | 28% |
| 21–40 | 16 | 30% |
| 41 to 64 | 22 | 42% |
| Traditional Medicine Use | 33 | 61% |
| Used tourniquet | 16 | 30% |
| **Clinical Care** | | |
| Received antivenom | 22 | 42% |
| Reached allopathic healthcare facility | 26 | 48% |
| Underdose (n = 22) | 16 | 72.72% |

## Therapeutic itineraries

Therapeutic itineraries varied significantly as snakebite victims sought healing following their accident. The most common self-reported symptoms were pain, swelling of the wound site, dizziness, vomiting, and bleeding from the bite wound, eyes, teeth, and ears. Snakebite victims received help from family members, medical professionals, curanderos, neighbors and strangers. Assistance from family members, coworkers, neighbors and strangers included preparing and administering homeopathic remedies, carrying victims from the mountain or farm to town, transporting the victim to the hospital, and searching for antivenom (Fig 4).

The therapeutic itinerary begins with the snakebite incident. Based on our analysis of case history interviews, most interviewees experienced envenoming, and thus risk of sickness, permanent injury or death. Yet, there is potential for confusion in self-diagnosis at the time of the

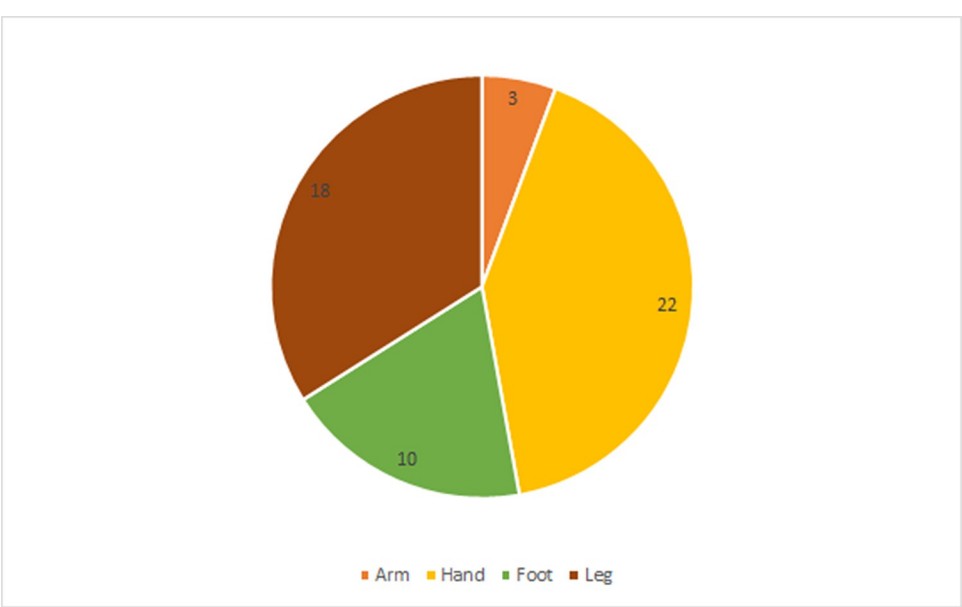

**Fig 2. Anatomical site affected in reported snakebite incidents.**

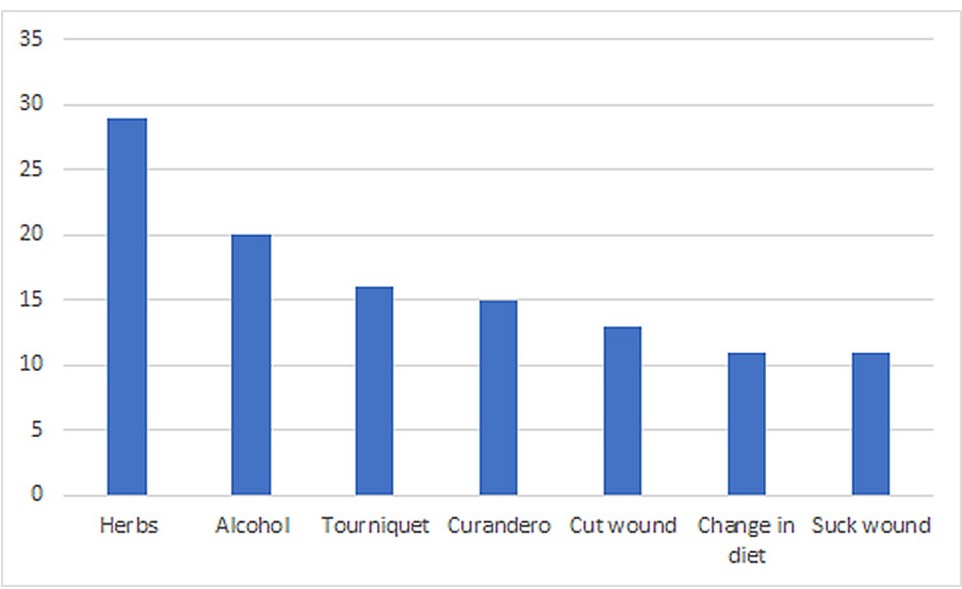

**Fig 3. Types of traditional remedies used, among 33 respondents who reported any traditional medicine use.**

bite, given reports of mild or asymptomatic snakebites. For one, some venomous and non-venomous snakes look similar. The cotorrera (*Bothriechis bicolor*) was the most reported species involved in snakebite incidents, as this snake makes its home in high-elevation coffee trees in

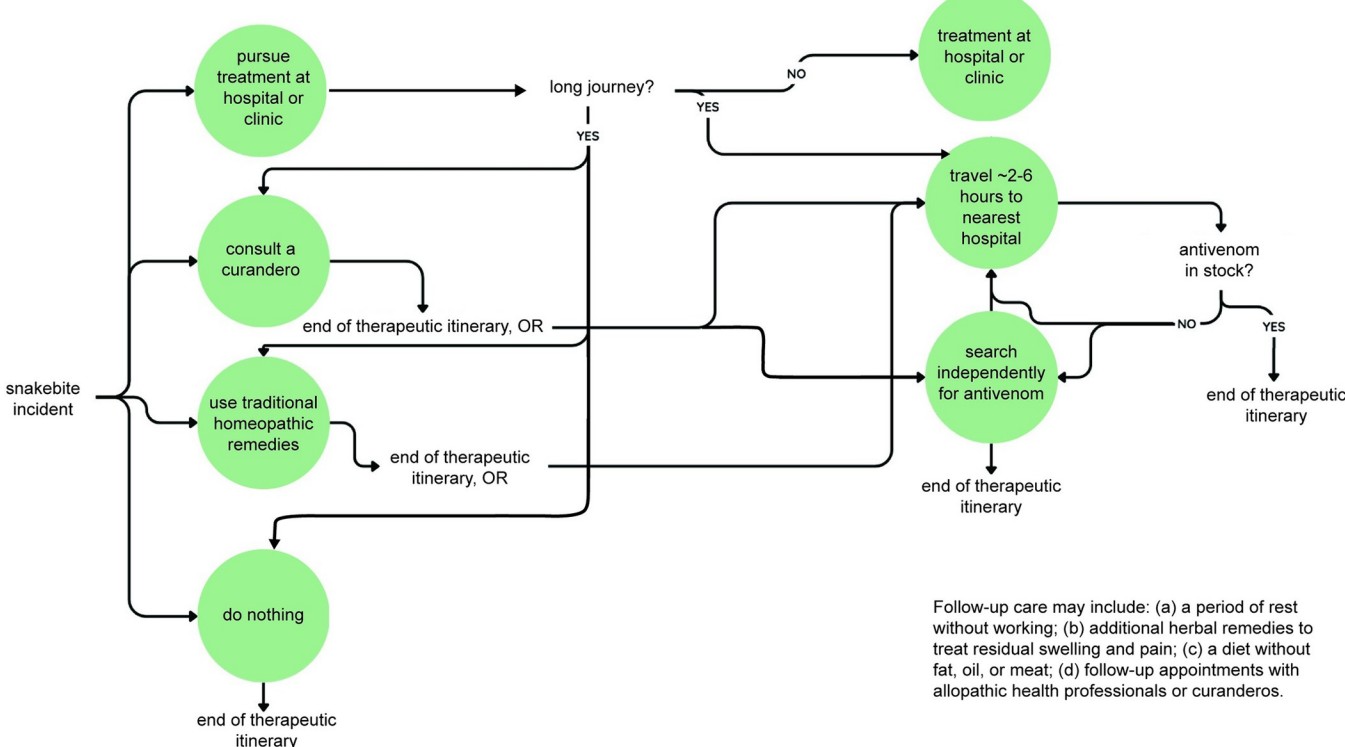

**Fig 4. Schematic diagram of therapeutic itineraries, starting with snakebite incident (by authors).**

Chiapas. This snake has a bright green color with teal spots and mild venom, causing mostly symptoms of pain and local swelling [29]. In the Sierra Madre region, there is also a similar-looking snake that is entirely non-venomous. The harmless bejuquilla, or *Oxybelis fulgidus*, shares the cotorrera's habitat and bright green color. The similarity of these two species is a potential cause of confusion, and respondents mentioned changes in health-seeking behavior as a result of perceived asymptomatic cotorrera bites, which may have been caused by the bejuquilla.

"*There are two types of these green snakes, there is one that is green green, and another that has light blue spots. They say that this one with spots is more dangerous, many have been bitten. You don't die, but it causes a lot of pain. But Jose resisted, I don't know if he is stronger.*"—Jose's father (Jose bitten 2021).

Two victims mentioned that a bite could be less severe if only one fang penetrated the wound. Others mentioned luck or personal "strength" when comparing their asymptomatic experiences with those of peers, who reacted to envenoming from what they thought was the same species.

"*Another friend was bitten [by the same snake], and I didn't pay attention to him because [that snake] had already bitten me in the past. But he swelled up, he was resting for about 5, 6 days. . . There are people with stronger blood, like me, who are not affected by the snake.*" Luis, bitten 3 times in 1978, 1987, and 1993.

To treat snakebite, interview participants tended to prefer allopathic medicine; however, traditional medicine was generally more accessible for economic, geographic, and social reasons. Hospitals were far from the site of most snakebites, so that victims had to travel hours to reach a medical facility. Distance seemed to be the most pressing obstacle to antivenom access (Fig 5). For those who remembered their travel itinerary (n = 24), respondents reported an average travel time of 2.76 hours just to reach the first medical facility.

"*Another guy here died. He grew ginger and was bitten by a viper. He didn't make it down [the mountain]. He was 32 years old, and left 3 kids and his wife. He died horribly.*" Daniela, bitten in 2013.

"*I cured it with a vine called* curarina. *I took it crushed, with alcohol. At this time there was no clinic, there was no hospital, there was nothing. . . If a snake bit me now, I would go to the municipality to the hospital, where they have medicine. Now there are doctors. Before, how? There was just* monte [mountain and/or bush]. *Those herbs are not enough.*" Arturo, bitten 1983.

"*I could barely bear the journey back to the town. It's two hours walking, and I came quickly. I was shaky. I said, 'I'm going to die here'*" Martin, bitten 2022 (Fig 6).

Even for those who reached the medical facility successfully, antivenom was rarely in stock. Those who reached a clinic or hospital (n = 22) received antivenom 23% of the time. Of the 22 victims who reached antivenom (either at the hospital, or independently), 16 reported receiving less than a minimum dose (5 vials for Antivipmyn), or reported that the hospital ran out of antivenom during treatment.

"*It's not easy to buy the antidote. It's hard to find. There is none.*" Lorenzo, bitten 2005.

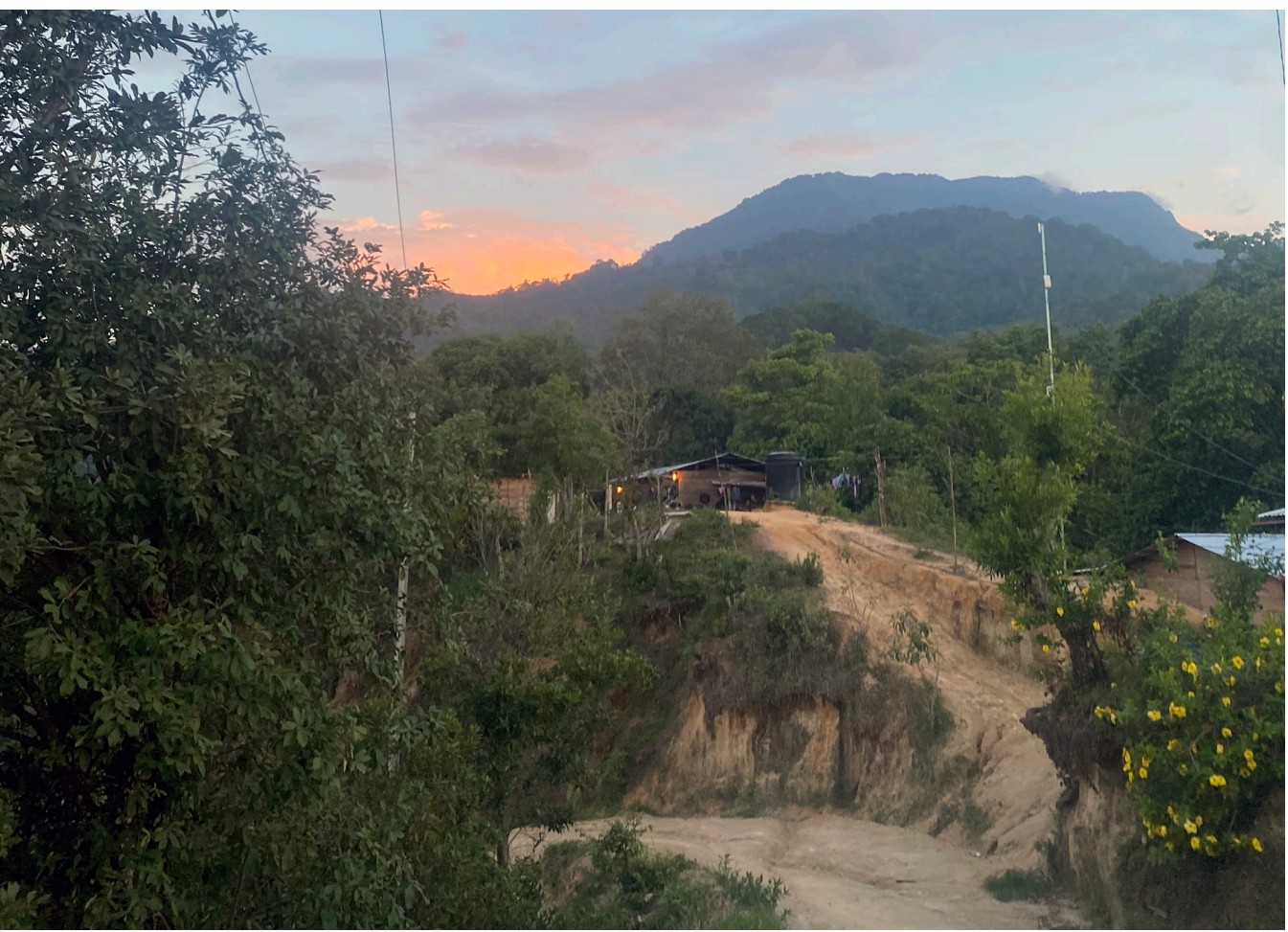

**Fig 5. Remote farm house in the Sierra Madre region of Chiapas (photo by authors).**

*"When we got to the hospital, they had the dose but it wasn't a complete one. It wasn't enough"* Marco, bitten 2015.

"*They got to the hospital 30 minutes later, but the hospital didn't have any antivenom. Silvester's family went to look for the antivenom and they bought two from a private doctor for 3,500 pesos* [$200] *each. They wanted to buy more, but the doctor only had two.*" Field Notes on Silvestre, bitten 2021.

Cost was another obstacle to receiving antivenom. Of the 22 patients who received antivenom, 12 had to pay out of pocket for at least one vial. When discussing their therapeutic itineraries, many respondents mentioned the costs of transportation and accessing antivenom. In the case of Federico, bitten 2008, hospital authorities expected payment for the ambulance trip. Federico's brother had to seek funds from a campaigning politician, who helped them if they promised not to consult a curandero. This caused a delay in transportation to the next medical facility, where Federico could be treated.

*"I was cleaning [weeding] coffee [plants] when it [the snake] bit me. My son-in-law did me the favor of bringing me to the hospital. When we got to Mapastepec, there was no medicine,*

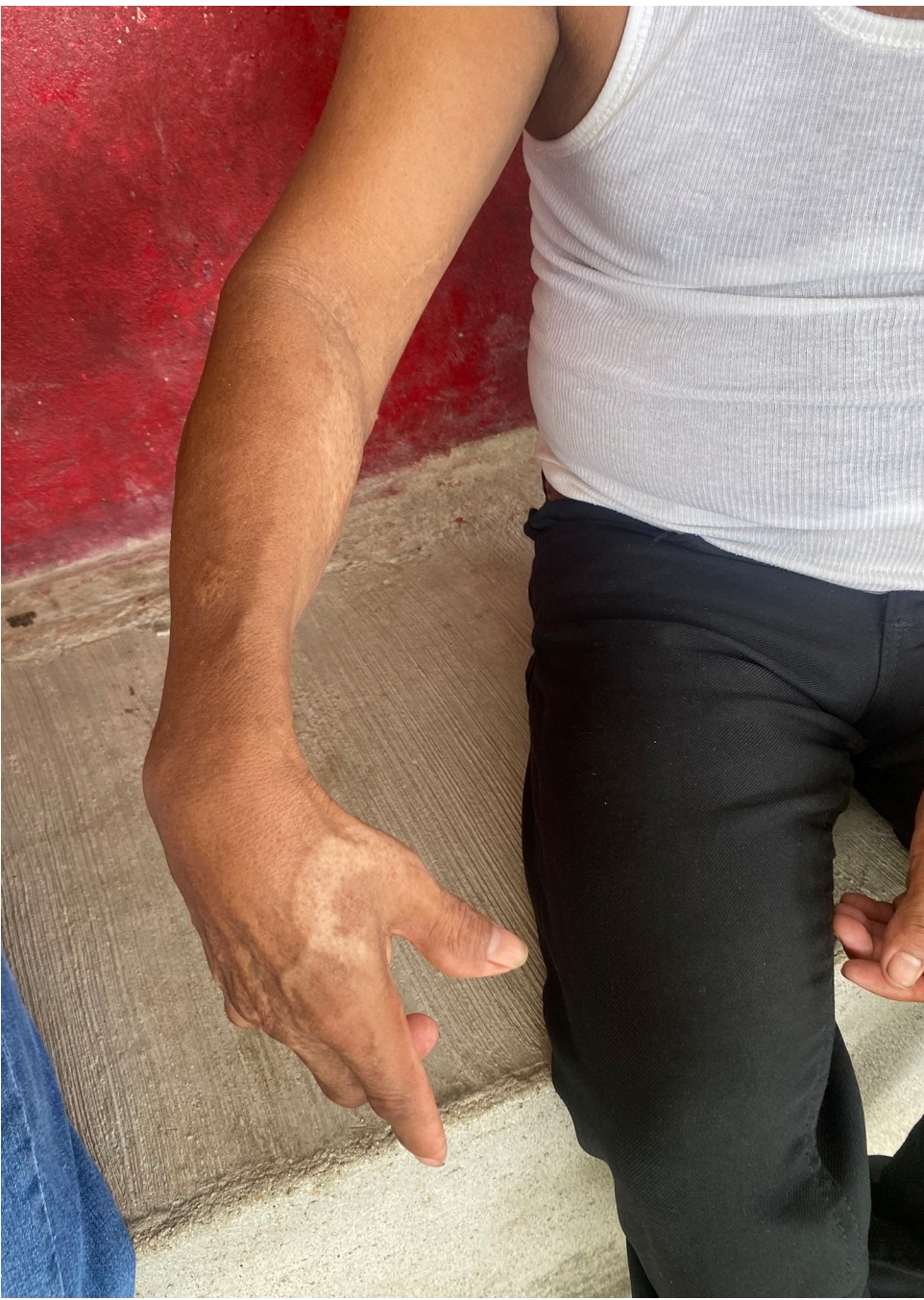

**Fig 6. Martin, showing the site of his snakebite wound (photo by authors).**

*nothing. That's when my breathing stopped. They transferred me to Tapachula. When we got there, there was no medicine there either. They told me that they had one in San Cristobal [the state capital], but the doctor said, 'by the time the medicine gets here, you'll be dead.' Well, we had to check around there [in Tapachula]. My family went to the veterinary for animals. They found a vial there. But they're expensive. 3,500 pesos [$200] per vial. They injected me with around 5."* Eduardo, bitten 2019.

"*We took him immediately to the nurse here, and she injected him with a small vial. But it's expensive. . . We injected one vial here so he would survive until La Revolución, and from La Revolución to Villa Flores because they didn't accept him in La Revolución.*" Interview with Emilio's wife, Emilio bitten 2016.

"*I think that they should have the antivenom in all the clinics. As the health center here is very small, we don't have doctors. When things are serious, we go to Valle or Tuxtepec. And at the very least you have to have money, because if not, they won't treat you. The snake antivenom is expensive.*" Daniela, bitten 2013.

Consulting clinics or hospitals did not exclude the possibility of a later visit to a curandero, or vice versa. Many victims who chose to seek medical help reported using traditional medicine or visiting a curandero to survive the long journey to medical facilities. Upon arrival at the hospital, most victims reported that antivenoms were out of stock. At this point, the victim could risk the journey to the next hospital (usually 2–4 hours by car or ambulance). Alternatively, the victim's friends or family could go search for antivenoms in pharmacies or through individual health practitioners. On these occasions, families had to pay out of pocket for the antivenom, reporting a price around 3,500 pesos (US$200) per vial, with a recommended dose starting at 5 vials.

There are other extraneous reasons why a victim would have a negative perception of biomedicine. Victims mentioned the secondary effects of Western medicine (n = 1), a fear of amputation (n = 1), and a fear of fasciotomy, an operation where surgeons slice the limb in order to relieve excessive pressure caused by swelling from a snakebite (n = 1).

There were two other common paths. One was a purely homeopathic solution, which was most common among victims who reported few severe symptoms (e.g., did not experience bleeding from eyes, ears, or gums), or in cases where medical facilities were perceived as out of reach or nonexistent. Another important therapeutic itinerary involved the independent search for antivenoms. Victims or families purchased antivenoms from local nurses or medical professionals, pharmacies, or even, in one case, from vector-control brigades. On one occasion, a nurse claimed the hospital had no antivenom. After the victim's family searched fruitlessly for the antivenom and returned empty-handed, a doctor scolded the nurse and affirmed that the hospital had antivenom. Multiple accounts reported nurses and doctors selling the antivenom privately.

Of the 22 victims who reached antivenom, 72.7% (n = 16) reported receiving a dose less than 5 vials (the minimum recommended dose of Antivipmyn), or reported that the doctors wanted to inject more vials but ran out of antivenom. We counted these as underdosages. These included victims who purchased antivenom from outside of hospitals. Estimated time to reach hospitals and antivenom are based on victims' estimations. These estimates do not consider those who could not recall the amount of time spent in transit.

Many victims who were bitten over two decades ago mentioned that new local clinics and hospitals make healthcare more accessible than before. To analyze change over time and mitigate the problem of recall bias, we compared treatment-seeking behaviors in two time periods, 1977–2010 and 2011–2023 (Fig 7). More recent victims were somewhat more likely to reach hospitals and to receive antivenom, and less likely to have used traditional medicine. Despite victims' reports of improved hospital access, recent victims reported longer travel times to hospitals and to reach antivenom, as compared to pre-2011 victims (Table 2). However, we note that many of the people we interviewed could not recall the time-related information at all, especially in the pre-2011 group.

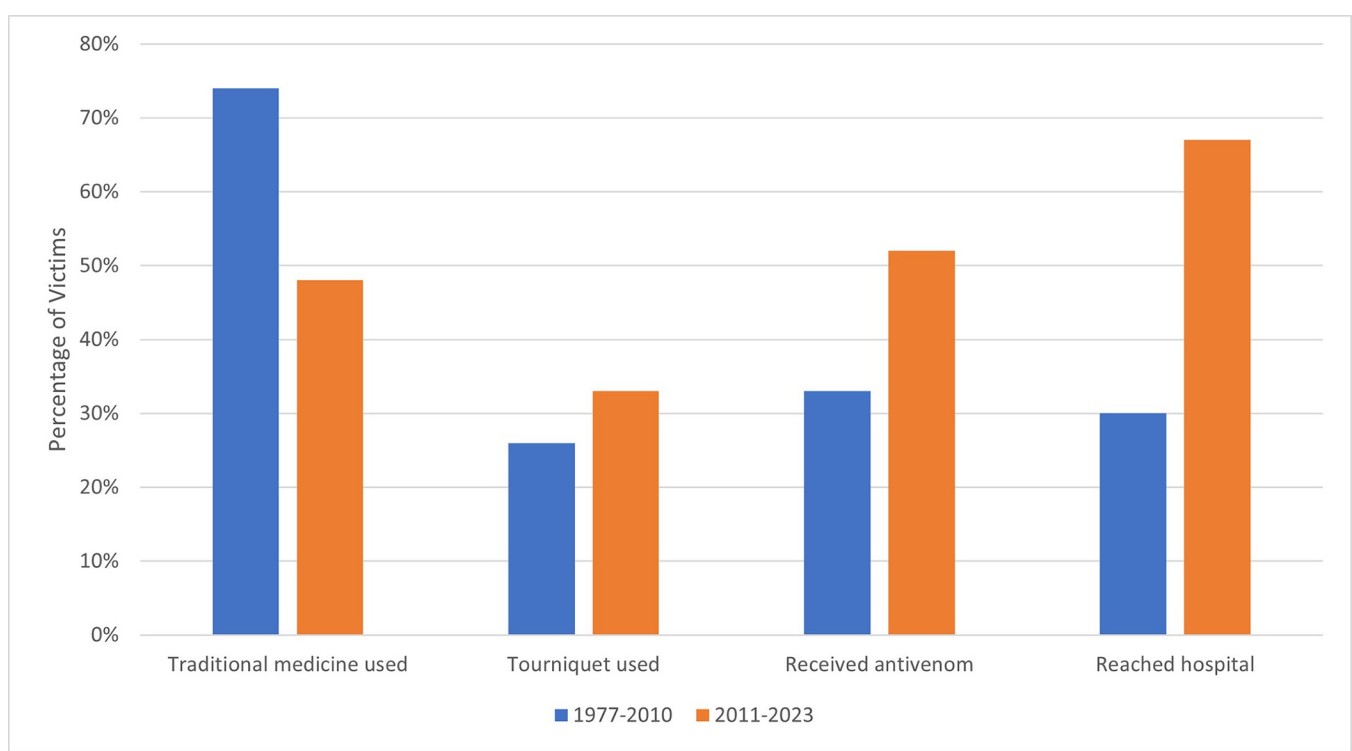

**Fig 7. Comparison of treatment-seeking behaviors for two time periods, 1977–2010 and 2011–2023.** Percentage of incidents shown. There are 27 incidents in each group.

## Rationales for use of traditional remedies

Often, interviewees sought both traditional and biomedical treatments following a snakebite. Victims used herbs, alcohol, or consulted traditional doctors as a replacement for medical care, or in order to survive the journey to the hospital.

"*As we hurry to get [the victim] to the hospital, we use homeopathic remedies.*" Carlos, bitten 2021.

"*First I went to [the curandero] to survive the journey to the hospital in Jaltenango, because the journey is very long.*" Gabriel, bitten 2023.

"*If the snake is large, you'll die before reaching the hospital. So, it's better to stay put and not move. . . Chili and alcohol are the best cures for neutralizing venom. That's what saves people's lives here, because there's not enough time to reach the hospital."* Agustin, bitten in 2021.

General health knowledge and folk remedies in the study region seemed driven, directly or indirectly, by the hot/cold humoral system. While only a few interviewees explicitly explained

**Table 2. Average time elapsed to reach hospital and to access antivenom.**

| | All | | | 1977–2010 | | | 2011–2023 | | |
|---|---|---|---|---|---|---|---|---|---|
| | N | Mean (hours) | SD | N | Mean (hours) | SD | N | Mean (hours) | SD |
| Time to reach hospital | 24 | 2.76 | 4.673 | 7 | 1.26 | 0.72 | 17 | 3.38 | 5.46 |
| Time to reach antivenom | 21 | 3.607 | 4.028 | 8 | 2.07 | 1.56 | 13 | 4.55 | 4.81 |

the hot/cold cosmovisions of medicine, most of the homeopathic remedies reported have been documented as "hot" cures for "cold" snakebites, or otherwise mentioned by other interviewees as a "hot" cure. The most common "hot" cures included alcohol consumption (n = 20), steam baths (n = 4), chili (n = 4), garlic (n = 4), lime (n = 2), gasoline (n = 2), hot water (n = 2), or a heated machete (n = 1). Victims who subscribed to hot/cold cosmology mentioned the "cold" nature of snake venom, and some suggested avoiding water following a snakebite.

"*I blended the chili and ate it. I had garlic, I chewed it up as well because garlic is hot. In the case of the snake, which is cold, you have to take warm things. Chili is also good because it is hot.*" Francisco, bitten 2023.

"*If you drink water, death from snakebite is quick, they say.*" Adrian's father, Adrian bitten 2020.

"*When you are bitten by a snake you should not get wet, it is bad.*" Enrique, bitten 2022.

At least one person suggested that conventional medicine offers support for beliefs based on the hot/cold system:

"*The doctor who treated me told me that I was lucky to have lived, because the cold of the river water hit me.*" Matias, bitten 2017.

Practices of traditional healers vary somewhat between the two study areas. In the Sierra Madre, victims consulted curanderos (who reportedly used herbs, cutting, sucking, and alcohol) and yerberos (who used a mix of herbs and pharmaceutical medicines). In Oaxaca, all interviewees in the Chinantla Baja region were familiar with one "culebrero" (i.e., traditional snakebite healer) called Don Pepe, who would use a mix of spiritual rituals (prayers), herbs, a "ventosa" (fire cupping) suction practice, and steam baths to heal snakebite victims. These methods matched reports of "hot" treatments for "cold" snake envenoming from other sources [19,21]. According to an interview with a local biomedical nurse, Don Pepe had a tattoo of the devil on one arm and the Virgin Mary on the other. In order to become a culebrero, Don Pepe underwent a ritual in the forest, during which snakes of every size slithered up his legs and around his head, and he had to kiss them. Since Don Pepe's death, the continuation of the culebrero practice in Chinantla Baja is in doubt.

In both Oaxaca and Chiapas, curanderos or people described as "knowing about herbs or medicine" were generally more accessible than clinics and had a reputation among the communities they served. In some cases, victims reported that these traditional medicine doctors collaborated with biomedical health professionals. Victims reported that Don Pepe worked with a nurse who inspected victims for signs of necrosis, and injected "vitaminas".

"*Don Pepe took the venom out of me, and he was working with another doctor who gave me vitamins. The viper's venom burned my blood, so they gave me vitamins. . . [Don Pepe] gave me a steam bath. He put my foot on top of a log, covered it up, heated a stone and made [the wound site] sweat with the steam.*" Antonio, bitten in 2008, 2010 and 2013.

We documented no contradictions between traditional cosmovision and an acceptance of allopathic medicines or treatment, with the notable exception of one victim who claimed that "suero" (saline solution) was cold, and therefore dangerous for snakebite victims because they were already suffering the cold effects of venom. The same victim drank alcohol before visiting the hospital, and believed that this had neutralized the venom.

"*With pure alcohol the venom was neutralized. They gave me serum, but. . . I didn't want them to give me the serum because the serum is cold and the snake venom is cold too. So alcohol helps the poison more because it is hot. And alcohol cuts the venom, because they did 5 tests on me and there was never any snake venom.*" Mario, bitten 2016.

Others had poor experiences with biomedical health facilities, due to antivenom shortages, absenteeism, long wait times, or a refusal to attend victims.

"*They took me to Tuxtepec but [the doctors] didn't want to treat me. They said that the venom had already been in me a long time.*" Daniela, bitten 2013.

"*They sent me to [another hospital] in Villa Flores but in Villa Flores they did not treat me because they said it was not a dangerous animal [that bit me]. I said that it was a dangerous animal, that my mouth was bleeding. That doctor told me no, that I was very dirty and that's why my mouth was bleeding. And they didn't believe me. . . They transferred me again to Revolución, and my mouth began to bleed more but they didn't treat me there, and they discharged me in Revolución. My mouth continued to bleed. I was angry.*" Juan Carlos, bitten 2021.

"*Miguel Angel went to the [local] clinic but there was no one there. The brother went to get the antivenom. Miguel Angel's brother found a bottle of antivenom 2 hours later, on a family ranch near Santa Rita.*" Field notes on Miguel Angel, bitten 2008.

## Discussion

Given poor antivenom availability in remote rural areas, victims were forced to make a difficult decision. They could undertake a long trip to the hospital, unsure of whether they would survive the painful journey and not knowing how much this medical care would cost them. Or, they could remain in their community among their loved ones, and hope to survive the venom's effects. This decision was further complicated by victims' ambivalent perceptions of allopathic medicine, based on prior experience. While interviewees strongly believed in the effectiveness of antivenoms, many reported problems with the medical system.

The key barriers to antivenom access can be summarized as distance, cost, and unreliable supply, similar to what has been documented in the literature around the world [3,7,8]. The largest financial burden for victims was antivenom, for those who found it. The drug costs around 3,500 pesos (US$200) per vial, according to interviews, and a minimum recommended dose for Antivipmyn is 5 vials [30]. Victims suffer not only the debts associated with their healthcare, but also the foregone income as they recover and deal with often permanent disability [33]. The cost of transportation to medical services can also be high, and seeking funds for transportation and antivenom can lead to additional delays in treatment.

Even after overcoming barriers of distance and cost, victims faced unreliable antivenom availability, even at hospitals. By and large, victims knew about antivenom, considered it effective, and wanted to access it; many mentioned antivenom by name, or mentioned a familiarity with the "medicine against snakes" from before their incident. When recounting their therapeutic itineraries, victims repeatedly expressed frustration that antivenom was not available and accessible. A few patients reported frustration over hours of waiting while doctors ran tests to check for envenoming, and the patients might not be notified if the hospital had no antivenom. Such negative experiences of the medical system—which have little to do with the effectiveness of antivenom—could reduce overall trust in allopathic care and decrease the likelihood that victims will make the journey to a hospital.

The context of the Mexican healthcare system should be considered. The hospitals available to rural snakebite victims in Oaxaca and Chiapas are, almost without exception, public hospitals. In Mexico, the public healthcare system is mostly decentralized to the different states, and quality is variable; southern states like Oaxaca and Chiapas rank lowest in the country in the HAQ (Health Access and Quality) index, although investment in the public health system of those states has also increased, and health outcomes improved, in recent years [34]. Each state is responsible for purchasing antivenom for its own region, and these are then distributed to various hospitals. Typically, antivenoms are supplied to larger hospitals rather than smaller ones, so patients often need to travel to access them. In public hospitals, antivenoms and treatment are provided free of charge, but in places where antivenoms are not available, individuals may purchase them on their own, from pharmacies or other suppliers, thus bearing the cost themselves.

Given the barriers to accessing antivenom in rural areas of Oaxaca and Chiapas, it is not surprising that victims turn to alternative medicine. While much past research views traditional medicine use as a cause of delayed antivenom access [7,8] or an obstacle to allopathic medicine use [8,31], we find that causality may also flow in the opposite direction: many snakebite victims in Southern Mexico reported using traditional medicine because of barriers to allopathic treatment, particularly the long travel required to reach adequate services, or a lack of trust in allopathic medicine offered in government health facilities, which corroborates research findings from around the world [14–16,33,35]. Traditional remedies also provide relief from the pain and thus make the often-long journeys to biomedical health facilities more tolerable, as found in a study in the Amazon basin of Brazil [7].

Our findings underscore the importance of an intercultural approach to promote collaborations and mutual understanding between traditional healers and personnel in official public health institutions. While we cannot make an evaluation of efficacy of homeopathic or herbal treatments [7,10,36], it is clear that the participants in our study see the value of both biomedical and traditional remedies, and, as in many places where there are competing medical models or frameworks, people choose therapies for practical reasons and depending on the nature of the problem [9,23,25]. The tendency to mix allopathic and traditional treatments for snakebite is consistent with other studies [7,16]. In our study area, victims prefer the standard biomedical treatment (antivenom therapy), but often find it out of reach. Many participants in our study doubt the efficacy of traditional medicine, even seeing such therapies as potentially dangerous. But there is a strong cultural anchor for herbs and other treatments connected to Meso-American "hot-cold" systems of folk etiology, with "hot" cures seen to counteract the "cold" of snakebite [18,19,21]. A robust and productive intercultural approach to snakebite should strive to understand the cultural foundations of health beliefs; analyze existing health-seeking behavior, especially as it combines allopathic and traditional practices; consider structural barriers to receiving care, such as poverty, poor transportation, or lack of healthcare resources; and view traditional healers as potential allies due to their social role as trusted messengers [9,23].

## Conclusion

This study of therapeutic itineraries of snakebite victims in southern Mexico confirms the widely held view that the main obstacles to antivenom access continue to be cost, distance, and scarce antivenom availability. Despite recent progress toward improved antivenom access in Mexico, snakebite victims continue to face high barriers to treatment and out of pocket costs. Standard antivenom treatment is highly valued and sought, even as traditional beliefs and practices persist; snakebite victims turn to traditional healers or homeopathic treatment

mainly to withstand the rigors of the long journey to treatment. Recent global analysis of vulnerability to snakebite [37] may underestimate the vulnerability of rural people in places like southern Mexico, to the extent that it does not consider the complexities of actually finding and obtaining antivenom treatment.

This study has certain limitations. We are unable to identify all species of snakes or herbal remedies used, because common names may refer to multiple species. While we consulted local biologists and herpetologists to minimize confusion, there is still much room for misidentification. Study participants were a convenience sample, based on recommendations from CONANP and victims' connections. Study sites were chosen as high-incidence areas for snakebite, and the authors make no claims about the epidemiology of SBE in the study region or outside. We note the risk of recall bias and memory erosion, which may make accounts of older snakebite incidents (including details about travel, symptoms, and treatment) less reliable. There may also be cognitive bias associated with mistaken self-diagnosis of envenomation, analogous to a placebo effect. There are two sources of this erroneous self-diagnosis: bites from non-venomous species and dry bites, when a venomous snake delivers a warning bite without venom [4]. Such situations of actually innocuous bites followed by treatment could lead victims to believe that they had been cured, when there was no venom to begin with [4]. While it is difficult to say with certainty, we believe that some of the snakebite victims we interviewed may draw false conclusions about the effectiveness of some traditional treatments due to such misunderstandings.

Today, snakebite victims in the Sierra Madre and Chinantla Baja regions must travel long distances over many hours to reach antivenom. In order to address these barriers to treatment, health ministries must improve antivenom stock such that patients know where they can find antivenom. Currently, there is no centralized system to track antivenom distribution in Mexico. A live inventory would lay the foundation for a more effective and efficient antivenom distribution and, subsequently, improved patient outcomes and faith in medical systems [38]. In addition, if there is a secondary market for antivenoms, a live inventory could improve oversight and reduce shrinkage. Besides improved inventory oversight, communities should have a clear plan of action in case of a snakebite. Clinics that transfer patients should, if possible, call the hospital ahead of time so that the recipients can prepare antivenom dosages or redirect the patient if they have no antivenom in stock. As suggested by Dr. Alfonso Suarez (pers. comm.), until antivenom distribution systems are improved, we recommend an antivenom distribution system where antivenom doses are stored in strategic locations across rural communities. Future interventions should aim to build trust between doctor and patient, across cultural and class boundaries, and between traditional healers and public health personnel. An intercultural health approach helps us understand the rationales for treatment-seeking within appropriate cultural contexts without losing sight of structural obstacles to receiving adequate care.

## Supporting information

**S1 Dataset. Dataset for quantitative analysis.**
(XLSX)

**S1 Interview summaries. Interview summaries.**
(ZIP)

## Acknowledgments

We thank Miriam Yanet and Ismael Galvez from the CONANP, the National Commission for Protected Areas, for helping to arrange field work; Pablo Quintana Ahuja for his assistance in

the field; Dr. Alfonso Suarez Velazquez, Dr. Luis Pena Garcia, Dr. Roberto Luna Reyes, Enrique Sandoval Orozco, and Jordan Edgardo Bermúdez Casillas, for their technical advice; Paul Dosh, Xavier Haro-Carrión, and Dan Trudeau of Macalester College for help with methods; and Louann Terveer from the Macalester College library for publishing advice. Research was supported by a Mann-Hill Summer Research Fellowship from Macalester College.

## Author Contributions

**Conceptualization:** Chloe Vasquez, Eric D. Carter.

**Data curation:** Chloe Vasquez, Eric D. Carter.

**Formal analysis:** Chloe Vasquez, Edgar Neri Castro, Eric D. Carter.

**Funding acquisition:** Chloe Vasquez, Eric D. Carter.

**Investigation:** Chloe Vasquez, Edgar Neri Castro, Eric D. Carter.

**Methodology:** Chloe Vasquez, Eric D. Carter.

**Project administration:** Chloe Vasquez, Eric D. Carter.

**Resources:** Chloe Vasquez, Edgar Neri Castro, Eric D. Carter.

**Supervision:** Chloe Vasquez, Eric D. Carter.

**Validation:** Chloe Vasquez, Edgar Neri Castro, Eric D. Carter.

**Visualization:** Chloe Vasquez, Edgar Neri Castro, Eric D. Carter.

**Writing – original draft:** Chloe Vasquez, Eric D. Carter.

**Writing – review & editing:** Chloe Vasquez, Edgar Neri Castro, Eric D. Carter.

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
