## [Decision Letter · Decision Letter 0]

25 Apr 2024

Dear Dr Carter,

Thank you very much for submitting your manuscript "Therapeutic itineraries of snakebite victims and antivenom access in southern Mexico" for consideration at PLOS Neglected Tropical Diseases. As with all papers reviewed by the journal, your manuscript was reviewed by members of the editorial board and by several independent reviewers. The reviewers appreciated the attention to an important topic. Based on the reviews, we are likely to accept this manuscript for publication, providing that you modify the manuscript according to the review recommendations. 

Sincerely,

Marco Aurélio Sartim

Guest Editor

José María Gutiérrez

Section Editor

Reviewer's Responses to Questions

**Key Review Criteria Required for Acceptance?**

**Methods**

-Are the objectives of the study clearly articulated with a clear testable hypothesis stated?

-Is the study design appropriate to address the stated objectives?

-Is the population clearly described and appropriate for the hypothesis being tested?

-Is the sample size sufficient to ensure adequate power to address the hypothesis being tested?

-Were correct statistical analysis used to support conclusions?

-Are there concerns about ethical or regulatory requirements being met?

Reviewer #1: This study is descriptive and does not aim to prove or disprove a specific hypothesis. In light off this, I believe it is of qualitative interest to understand the hurdles that snakebite victims face in the study region.

Reviewer #2: This is a funny but well-articulated manuscript that discusses many important considerations in snake envenomation not only in rural Mexico but also worldwide, particularly in rural South America, Africa, and Asia.

Reviewer #3: The hot and cold system therapeutic framework has been described to explain practices around snakebites. In this way, some additional data of those living in Sierra Madre and Chinantla Baja regions would be useful to understand their cultural or anthropological background. Was there any difference in the perceptions of snakebite risk, use of traditional medicine or confidence in antivenom treatment?

Interviewing victims of snakebites in their lifetime might cause recall bias. Authors noted the risk of cognitive bias associated with mistaken self-diagnosis of snakebite. However, no there were comments regarding the possibility of distorted or inaccurate memory of their therapeutic itineraries. In which extension, their responses might not accurately represent the reality? Has any strategy been implemented to lessen its effects on data accuracy to mitigate recall bias in the reports?

**Results**

-Does the analysis presented match the analysis plan?

-Are the results clearly and completely presented?

-Are the figures (Tables, Images) of sufficient quality for clarity?

Reviewer #1: Yes, I only suggest presenting some of the results as figures, instead of in-line numerical descriptions.

Reviewer #2: Yes, they are.

Reviewer #3: As observed in other studies, the storytelling methods provided a very illustrative scenario of the difficulties to the accessibility of antivenom. Tables and photos are sufficiently clear and contributed for a better understanding of the results.

Distance was referred as a barrier to antivenom access, but the mean time to reach a hospital was less of three hours (although considerable standard deviation). If better roads and improvements in the public health system might be associated with reduction of the distance, in which proportion other variables contributed to the perception of barriers in accessing antivenom? It is not clear if the use of traditional medicine at the community would be considered a cause or consequence of the limitation of transportation for the time to reach hospital. 

There is no information regarding the health policy in the country and how antivenom is distributed. Should antivenom treatment be always paid by the patient? Does it mean the public health system in Mexico does not cover antivenom treatment? Please, convert price of antivenom (3,500 pesos per vial) from Mexican currency to American dollar.

**Conclusions**

-Are the conclusions supported by the data presented?

-Are the limitations of analysis clearly described?

-Do the authors discuss how these data can be helpful to advance our understanding of the topic under study?

-Is public health relevance addressed?

Reviewer #1: I am not a social scientist, so as far as my understanding of the study I don't have any major concerns.

Reviewer #2: Yes, they are.

Reviewer #3: Some more discussion is lacking about the confidence in the antivenom treatment, which was noticeable, and with no conflict with traditional medicine. 

Also, could the hot and cold theory contribute to the recognition of the relevance and intercultural applicability of these medicines, both in the clinical, academic, and political settings?

**Editorial and Data Presentation Modifications?**

Reviewer #1: Comments on: Therapeutic itineraries of snakebite victims

I found the acrticle very well written and a very valuable descriptive study. I feel however that the findings, ranging from qualititive to purely quantitative would benefit from putting into a wider context (see Longbottttom et al. 2018. The Lancet. https://doi.org/10.1016/S0140-6736(18)31224-8).

The results section, Characteristics of snakebite victims and Post-bite practices, would benefit from including some graphics, especially those which highlight the study's aims such as frequency of treatments sought. Pie charts would help communicate these findings.

Reviewer #2: Acknowledgments:

Line 561. "I also thank", who is I?

Line 566. "who accompanied me", who is me?

Reviewer #3: Line 23 – Add neuroparalytic effects when describing the variety of symptoms and associated deaths, as elapids maybe also a cause of morbidity and mortality. 

Line 112 – Metlapilcoatlus occiduus should be typed in italic.

**Summary and General Comments**

Reviewer #1: The main weakness is the small sample size. They make up for this with a lot of information from each interviewee, but some comparison of experiences between study sites may be of value to understand how the small sample size may affect their conclusions.

Reviewer #2: I think is an interesting manuscript and has value to be published.

Reviewer #3: There is a tendency to use qualitative methods to collect data and discuss the results in studies referring snakebite envenoming in traditional groups, as this interesting study of the therapeutic itineraries of patients bitten by snakes to reach antivenom treatment in Southern Mexico.

PLOS authors have the option to publish the peer review history of their article (what does this mean?). If published, this will include your full peer review and any attached files.

Reviewer #1: No

Reviewer #2: Yes: Alejandro Alagon

Reviewer #3: No

Figure Files:

Data Requirements:

Reproducibility:

References

---

## [Decision Letter · Decision Letter 1]

19 Jun 2024

Dear Dr Carter,

We are pleased to inform you that your manuscript 'Therapeutic itineraries of snakebite victims and antivenom access in southern Mexico' has been provisionally accepted for publication in PLOS Neglected Tropical Diseases.

Best regards,

Marco Aurélio Sartim

Guest Editor

José María Gutiérrez

Section Editor

Reviewer's Responses to Questions

**Key Review Criteria Required for Acceptance?**

**Methods**

-Are the objectives of the study clearly articulated with a clear testable hypothesis stated?

-Is the study design appropriate to address the stated objectives?

-Is the population clearly described and appropriate for the hypothesis being tested?

-Is the sample size sufficient to ensure adequate power to address the hypothesis being tested?

-Were correct statistical analysis used to support conclusions?

-Are there concerns about ethical or regulatory requirements being met?

Reviewer #1: Yes

Reviewer #2: The study's objectives are articulated, and despite the complexity of the survey, the populations are clearly described. The sample size is what it is, but sufficient to support the conclusions. I have no concerns about ethical or regulatory issues.

Reviewer #3: Yes

**Results**

-Does the analysis presented match the analysis plan?

-Are the results clearly and completely presented?

-Are the figures (Tables, Images) of sufficient quality for clarity?

Reviewer #1: Yes

Reviewer #2: The results follow the analysis plan and are presented. The figures have sufficient clarity.

Reviewer #3: Yes

**Conclusions**

-Are the conclusions supported by the data presented?

-Are the limitations of analysis clearly described?

-Do the authors discuss how these data can be helpful to advance our understanding of the topic under study?

-Is public health relevance addressed?

Reviewer #1: Yes

Reviewer #2: All of the above were well-addressed.

Reviewer #3: Yes

**Editorial and Data Presentation Modifications?**

Reviewer #1: No further comments

Reviewer #2: Accept

Reviewer #3: No more modifications are needed.

**Summary and General Comments**

Reviewer #1: The authors replied in sufficient detail to all reviewer comments. I believe this article is ready for publication.

Reviewer #2: This is a very interesting manuscript with lots of interdisciplinary information. It points out snake-envenomed patients' difficulties in finding proper treatment (antivenom). If published, it certainly will help the health authorities of Mexico to distribute antivenom more effectively. I think the conclusions achieved in the two high-incidence areas can be extrapolated to many other regions of Mexico and possibly other countries.

Reviewer #3: The revised manuscript included all the suggestions presented to the authors, either in the methodological aspects, the limitations of the study and in the issues of interculturality of the communities involved.

Important aspects about accessibility to antivenom treatment was provided, which makes the article an important contribution to the improvement of public policies for the distribution and availability of antivenoms.

PLOS authors have the option to publish the peer review history of their article (what does this mean?). If published, this will include your full peer review and any attached files.

Reviewer #1: No

Reviewer #2: **Yes: **Alejandro Alagón

Reviewer #3: No

---

## [Editor Report · Acceptance letter]

29 Jun 2024

Dear Dr. Carter,

We are delighted to inform you that your manuscript, "Therapeutic itineraries of snakebite victims and antivenom access in southern Mexico," has been formally accepted for publication in PLOS Neglected Tropical Diseases.

Best regards,

Shaden Kamhawi

co-Editor-in-Chief

Paul Brindley

co-Editor-in-Chief
